# Competencies for Agricultural Advisors in Innovation Support

**Charlotte Lybaert [1,\*], Lies Debruyne [1] , Eva Kyndt [2,3] and Fleur Marchand [1,4]**

1 Flanders Research Institute for Agriculture, Fisheries and Food (ILVO), 9820 Merelbeke, Belgium; lies.debruyne@ilvo.vlaanderen.be (L.D.); fleur.marchand@ilvo.vlaanderen.be (F.M.)

2 Department for Training and Education Sciences, University of Antwerp, 2000 Antwerpen, Belgium; eva.kyndt@uantwerpen.be

3 Centre for Transformative Innovation, Swinburne University of Technology, Melbourne 3122, Australia

4 Centre for Research on Environmental and Social Change, Institute of Environment and Sustainable Development, University of Antwerp, 2610 Wilrijk, Belgium

\* Correspondence: charlotte.lybaert@ilvo.vlaanderen.be

**Abstract:** The expectation that agricultural advisors will facilitate Interactive Innovation is accompanied by novel expectations for their competency profile. In addition to their traditional technical basis, advisors are now expected to organise multi-actor processes, facilitate learning, mediate conflict, etc. Innovation support services are inherently diverse. To date, no precise list of competencies required by agricultural advisors to support Interactive Innovation has been defined. To form the basis for a competency profile, we examine the competencies currently being expected from an agricultural advisor. This suggested profile, developed in the context of the H2020 i2connect project, is based on a literature review, semi-structured interviews with co-creation experts, and an online validation workshop. We explore five themes: (a) basic disposition and attitude, (b) content competence, (c) methodological competence, (d) organisational competence, and (e) reflection, learning, and personal development. In practice, the profile can be used as either a tool for setting up co-creation processes or as the foundation for the development of new training materials. We conclude with a recommendation to create teams of advisors rather than relying on individuals, as a team is more likely to comprise the diversity of required competencies.

**Keywords:** competencies; agricultural advisory services; multi-actor approach; co-creation; interactive innovation; innovation support

## 1. Introduction

Worldwide, the agricultural sector is being confronted with challenges such as climate change, biodiversity, and food security. Food systems account for one-third of global greenhouse gas emissions, consume large amounts of natural resources and do not allow for fair economic returns and livelihoods for all actors involved [1]. Furthermore, it has been widely recognised that food systems need to be sustainable to overcome crises, such as the COVID-19 pandemic [1,2]. The quest for innovative, sustainable solutions is vital to our short- and long-term survival. Innovations at the level of products, processes, and society are being sought in order to meet these challenges while addressing the needs as well as the possibilities of a range of agricultural companies. Agricultural advisory (extension) services provide support to farmers to help them find farm-level solutions for specific problems. Advisory services establish farmer–advisor service relationships to exchange knowledge and enhance skills [3]. A wide array of market and non-market entities that provide flows of information in agricultural sectors worldwide and agricultural advisory services are an important link [4]. The agricultural advisor, as the individual providing this specialised support to farmers, can play a number of roles. Advisors can be self-employed, public, or private sector employees. Regardless of their employment status, they can identify as technical experts, agents of the state, representatives of agri-businesses, change agents, etc. [5].

In recent years, more attention has gone to the role of advisors in the context of innovation processes. Agricultural innovation and advisory processes are recognised as being highly complex [6,7]. Innovation is seen as a collective process that involves multiple actors and diverse dynamics [7,8]. The EU, under the European Innovation Partnership Agricultural Productivity and Sustainability (EIP-AGRI), has promoted the 'Multi-Actor Approach' as the ideal model to reach innovation. The Multi-Actor Approach is a practical translation of Interactive Innovation, defined by the European Commission as farmers, farm advisors, scientists. and other stakeholders collaborating throughout a project to develop innovative solutions to practical problems, which then have a greater chance of being adopted [9,10]. This system thus requires an 'innovation intermediary' to facilitate these projects.

The EIP-AGRI designates agricultural advisors as these key innovation intermediaries. This is not surprising, considering the traditional importance of these individuals in the agricultural knowledge economy: They often have a close relationship with farmers and possess a good understanding of the local agricultural context. For many advisors, this novel function of innovation intermediary represents an extra role that must be fulfilled in addition to their other tasks. This new role also implies a change in the agricultural advisors' competency profile. Advisors are now expected to facilitate learning, play the role of educator, possess refined interpersonal and communication skills, resolve conflicts, etc. [11–14].

Agricultural literature on intermediary functions and innovation systems is growing [6, 15–17], together with literature about competencies for agricultural advisors [18–22]. To our knowledge, only limited research has been conducted on required competencies for agricultural advisors in the context of providing innovation support [23,24] with extensive lists of competencies, yet often without a clear explanation. The aim of the present study is to provide a more nuanced answer to the question, "Which competencies do agricultural advisors need to facilitate Interactive Innovation processes?".

First, we elaborate on the concepts of competence and competency and the intermediary role that is now expected of the advisor. Next, the framework used to explore the competencies required for innovation support is presented. Based upon the qualitative research resulting from the EU-funded i2connect project, we then present a competency profile for the innovation advisor (i.e., the agricultural advisor providing innovation support). We conclude by discussing the results and presenting opportunities for further research.

In this article, the term 'project' is used in its broadest sense, encompassing not only formally defined projects but also informal encounters and other processes.

## 2. Definition of Key Concepts

### 2.1. Competence and Competency

Competence and competency are concepts that have received a great deal of attention over the last decade in various fields of research. To avoid confusion about the terms 'competence' and 'competency', we define them here. Mulder (2001) defines competence as "the capability of a person or an organisation to reach specific achievements" [25]. A competency can be seen a part of competence, as competencies are characteristics of a person, team, or organisation which enable them to reach a specific achievement [25,26]. Personal competencies comprise an integrated set of performance-oriented capabilities [25]. These capabilities include clusters of knowledge structures: cognitive, interactive, psychomotor, and affective capabilities, as well as attitudes and values [25]. These concepts play a role in many practical contexts, e.g., Human Resource Management and professional development [25], and encourage scholars to think not only about knowledge itself but also about the knowledge required for competent work performance [27,28].

The most frequently used method of identifying a set of required competencies is job analysis, which describes personal competencies as specific sets of attributes that workers use to accomplish their work [27,29,30]. Three main approaches can further be distinguished, based on the way they identify competencies: a worker-oriented approach, a

work-oriented approach, and a multimethod-oriented approach [27]. The worker-oriented approach views competencies primarily as collections of attributes and personal traits possessed by workers, while the work-oriented approach takes the work as point of departure by identifying key activities and translating them into personal attributes [27,29]. In the present study, a multimethod-oriented approach was chosen as it comprises a more comprehensive approach to competence: It draws on principles from the aforementioned approaches, thus focusing on both work and worker [27,31]. It identifies competencies by looking at activities central to accomplishing specific work tasks, translates these into personal attributes, and identifies the personal characteristics (competencies) linked to effective job performance [31]. The following section presents an examination of the function of the innovation intermediary and investigates the competencies necessary to perform the required services.

*2.2. Roles and Functions of Innovation Intermediaries*

The role of the intermediary in an innovation network is called by many names: innovation broker, project monitor, translator, boundary spanner, etc. [32]. These innovation intermediaries perform a wide variety of tasks, all of which aim to collectively create knowledge that will ultimately lead to innovation. Many authors have conceptualised the function of the innovation intermediary in the context of agriculture, as well as in other scientific areas. For example, Klerkx & Leeuwis (2008) captured the main functions of an innovation intermediary in agricultural innovation systems under the headings 'demand articulation', 'network brokerage', and 'innovation process management' [16]. From another perspective, the Global Forum for Rural Advisory Services describes the role of extension and advisory services in agricultural innovation systems as being "about sharing and facilitating access to information, knowledge and expertise, and working with others to bring about innovation" [23]. This resonates with Hargadon & Sutton's definition (1997) of the role of brokers in the field of technology. Apart from a bridging function, they identify the role of brokers as a knowledge repository [33,34]. By dipping into this repository of existing ideas and combining them in novel ways, these brokers can offer their clients innovative solutions [33,34]. Another important framework is the work of Howells (2006), who studied the function of innovation intermediaries by conducting a set of case studies with managers in 22 UK-based organisations [34]. It became apparent that the case study organisations undertook significantly more functions than originally conceived [34]. Howells expanded the original list by what he calls 'unrecognised' and 'undervalued' functions [16,34]. The identified innovation intermediary functions can be summarised as (a) foresight and diagnostics, (b) scanning and information processing, (c) knowledge processing and combination/recombination, (d) gatekeeping and brokering, (e) testing and validation, (f) accreditation, (g) validation and regulation, (h) protecting the results, (i) commercialisation, and (j) evaluation of outcomes [34].

Some authors conceptualise the function of the innovation intermediary by defining different types of innovation support services (ISS). The view on ISS has changed in the 10 years from a linear communication model to a view of service provision as a learning process [8]. It is now generally agreed that ISS make innovation possible by fostering interaction and constructing knowledge [35]. Following Labarthe et al. (2013) and Faure et al. (2019) [35,36], this research considers ISS to be an activity rather than an organisational body. Several frameworks describe the functions of ISS [37,38]. One important typology is the 'revised generic ISS activities', presented by Faure et al. (2019) and based on the work of Mathé (2016) and Faure (2017) [39,40]. They defined seven generic ISS activities: (a) awareness and exchange of knowledge; (b) advisory, consultancy, and backstopping targeted activities; (c) demand articulation; (d) networks, facilitation, and brokerage services; (e) capacity building; (f) enhancing/supporting access to resources; and (g) institutional support for niche innovation and scaling mechanisms stimulation [35,39,40]. Faure et al. (2019) conducted research into the diversity of these ISS along different phases of the innovation process, based on the analysis of 57 case studies resulting from the

EU Agrispin project [35]. To describe the different stages of an innovation process, the 'Spiral of Innovations' model was used; this model distinguishes seven phases of an innovation process, starting with the initial idea to embedding the changed practices into the institutional environment [41]. The results show important insights in the diversity as well as the frequency of different types of ISS. For example, it appeared that 'Networking, facilitation and brokerage' services had the highest frequency count and were fairly even distributed over each phase [35]. In contrast, 'Enhancing/supporting access to resources' services appeared to be especially important in the planning and development phases [35]. These findings are relevant when determining the specific attributes required at different phases of the innovation process.

### 2.3. Qualifications of an Advisor Framework

The 'Qualifications of an advisor' framework was designed at the University of Hohenheim (Stuttgart) [42]. Their vision of advisory work is that advisors need to possess a specific attitude and personality, as well as expertise regarding content, methods, and management; they must also be willing to learn from experience and be able to reflect [42]. This vision resulted in five main themes: (a) basic disposition and attitude, (b) content competence, (c) methodological competence, (d) organisational competence, and (e) reflection, learning, and personal development (Figure 1). This framework was used to identify the competencies for agricultural advisors providing innovation support. The next section explains the methods used to construct the competency profile.

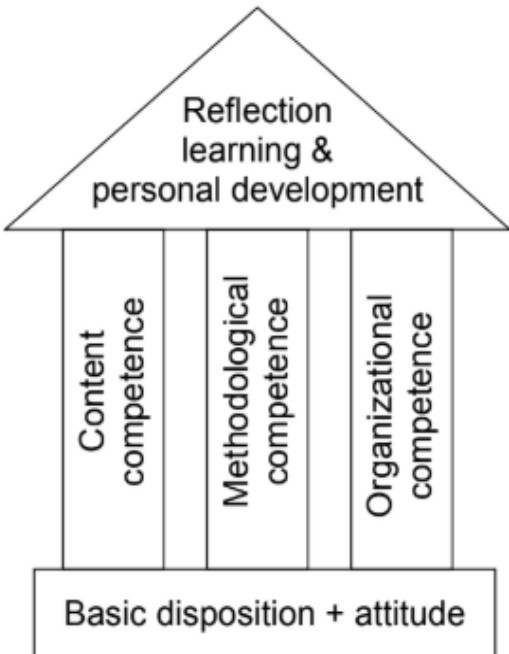

**Figure 1.** Qualifications of an advisor framework [42].

### 3. Materials and Methods

Identification of the competencies required for innovation support started with a thorough literature study. A search engine was used to simultaneously search a collection of scientific databases including Scopus, Science Direct, and Web of Science. Initial searches included a combination of the keywords "competence" or "competencies" and agricultural advisors, extension, brokerage, innovation (both in and outside the agricultural sector). The snowball method was used to find additional literature. All abstracts were screened and selected according to competencies for agricultural advisors or competencies for innovation support. A total of 61 articles were retained.

Seven in-person interviews were conducted with people that have significant experience in multi-actor projects; some but not all were part of the i2connect consortium. The experts were selected and invited for an interview upon recommendation by one of the i2connect consortium members. The two female and five male experts resided in Poland (2), the Netherlands (2), Austria (1), France (1), and Germany (1). These interviews were 'semi-structured', i.e., they consisted of conversations regarding the experts' experiences and opinions of activities for advisors in innovation processes and the relevant competencies needed to provide those services. The data collection reached a point of saturation after only a limited number of interviews.

The interviews were recorded and transcribed verbatim. Next, the competencies resulting from the literature review and the expert interviews were manually coded, using NVivo 12, according to the 'Qualifications of an advisor' framework [42]. The resulting coding tree can be found in Appendix A.

Subsequently, a validation workshop was organised with five members of the i2connect consortium. During this workshop, the five themes and their competencies were discussed based on their requirement for use in innovation support services. Then, the competencies were clustered in different groups: For each of the five themes of the 'Qualifications of an advisor' framework, a number of clusters were jointly identified with workshop participants on the basis of the competencies that were included in that theme. Some competencies were omitted when they were not deemed relevant or were already addressed under a similar competency. For example, one of the interviewees had identified 'mobility' (i.e., the ability to drive a car) as a required competency, while during the validation workshop, this competency was rejected because it was not deemed essential to innovation support.

The result led to the construction of the 'competency profile for the innovation advisor', discussed in the next section.

## 4. Results

The competency profile for the innovation advisor consists of five themes, following the structure 'Qualifications of an advisor' [42]. Each theme consists of a number of clusters, which in turn comprise several competencies. A summary of the competency profile for the innovation advisor is provided in Table 1. The interviews yielded largely similar competencies as those found in the literature study. The interviews differed from the literature, mainly by placing a greater emphasis on competencies belonging to the theme 'basic disposition and attitude', as well as including a greater number of competencies within the category of 'methodological competence'.

**Table 1.** Competency profile for the innovation advisor.

| **Basic Disposition and Attitude** | |
| --- | --- |
| Self-awareness | Self-awareness<br>Sense of equity<br>Willing to take a step back when needed<br>Willing to share power and give up control |
| Personal drive | Personal drive<br>Passion<br>Dedication<br>Trust in intuition |
| Sensitivity | Sensitivity<br>Responsiveness<br>Empathy<br>Emotional intelligence<br>Communication skills |

**Table 1.** *Cont.*

| | |
|---|---|
| Reliability | Reliability<br>Accountability<br>Trustworthiness<br>Ethics<br>Responsibility<br>Professional attitude |
| **Content Competence** | |
| Understanding the social context | Understanding the broader social environment<br>Connecting to the community<br>Understanding own role in the system<br>Being able to identify relevant actors |
| Understanding the Agricultural Knowledge and Innovation System (AKIS) | Understanding political and economic context<br>Basic knowledge about legal matters and the public policy of the region |
| Content knowledge | Background in agriculture<br>Technical knowledge<br>Ability to understand English |
| **Methodological Competence** | |
| Understanding the innovation process | Sensitivity for the process<br>Being able to recognise patterns in an innovation process<br>Knowing how to act in any given situation<br>Possessing and using tools related to innovation processes<br>Problem solving skills |
| Energy | Being able to keep energy and enthusiasm in the group<br>Being able to activate and mobilise people<br>Facilitation skills<br>Translation skills |
| Co-creation | Being able to identify crucial positions<br>Being able to identify missing positions<br>Good insight into human psychology |
| Mediation | Mediation skills |
| **Organisational Competence** | |
| Organisational competence | Planning<br>Meeting organisation<br>Following up with contacts<br>Keeping track of the network<br>Time management<br>Managing resources<br>Writing project proposals<br>Collecting funds<br>Delegating<br>Digital skills |
| **Reflection, Learning, and Personal Development** | |
| Reflection among peers | Habitually reflecting upon work with peers<br>Sharing a common language |
| Self-reflection | Habitually self-reflecting |
| Addressing professional network | Utilizing professional network |
| Lifelong learning aptitude | Ongoing skill development and learning<br>Knowing how to find new information |

### *4.1. Basic Disposition and Attitude*

Four clusters with competencies were identified under the theme 'basic disposition and attitude'. These competencies form the foundation of the advisors' competence [42]. The clusters are (a) self-awareness, (b) personal drive, (c) sensitivity, and (d) reliability.

### 4.1.1. Self-Awareness

The cluster 'self-awareness' regards self-knowledge. Innovation advisors must possess a sense of equity and should have an attitude which recognises the skills and knowledge of other actors. As one interviewee said: "*You are part of a network, you have your own competences, your own technical skills, but you are not better, or more important than the others. If you are there as* the *specialist, you block the road for co-creation.*" Important in this regard is an attitude of assistance and service, rather than leadership. Innovation advisors must be willing to take a step back when needed and must be willing to share power and give up control when the situation requires it. One of the interviewees shared her experience as an innovation advisor in operational groups: "*We try to be in the operational group when they have meetings. . . . And mostly it is very good to be there as guests. . . . We are viewing what happens and in case there are conflicts, we offer to moderate.*"

This cluster also touches upon trusting in one's own capability as well as the capability of others, which in turn relates to open-mindedness.

### 4.1.2. Personal Drive

The second cluster of competencies concerns motivation and personal drive. Innovation advisors should be able to evoke a certain passion for the project they serve and be dedicated to it. This requires an attitude of 'this is what I believe in' instead of 'this is what I have to do'. One interviewee referred to the concept of being a free actor: "*A free actor does things because he considers them necessary. Not because they are being directed or because they just want to make money. But because they are passionate about it. Think back to the teachers who left an impression during your education. They were enthusiastic people, with passion.*"

Furthermore, innovation advisors need to be able to trust in their own intuition. As one interviewee pointed out: "*Because intuitively you know more than you think.*"

### 4.1.3. Sensitivity

The cluster 'sensitivity' concerns the connection of the innovation advisor to other actors in the project. Innovation processes cannot be planned out in detail in advance. This requires the facilitator to be sensitive to what is occurring at the moment, to make a distinction between one situation and another, and thus be able to respond in an adequate manner. One interviewee commented: "*Sensitivity for what's happening is important. You need to be able to act without a plan.*"

This requires empathy and emotional intelligence, to be able to understand the needs of others and connect to them. For this, the innovation advisor needs to possess certain communication skills, such as the ability to listen, non-violent communication, and non-verbal communication skills.

### 4.1.4. Reliability

The fourth cluster of 'basic disposition and attitude' concerns the need to be reliable. The innovation advisor must be accountable for his/her actions and appear trustworthy in the eyes of the other actors. This relates to ethics, the common values of the workspace. These values will differ according to the context, as they are linked to a sociocultural background of the actors involved. Furthermore, the innovation advisor needs to be responsible and possess a professional attitude.

### *4.2. Content Competence*

For the theme 'content competence', three clusters were identified, which are linked to understanding the specific (agricultural) context the innovation process is embedded

in: (a) understanding the social context, (b) understanding the AKIS, and (c) content knowledge.

### 4.2.1. Understanding the Social Context

The first cluster concerns understanding the broader social environment in which the Interactive Innovation process is embedded. The innovation advisor needs to understand who the main actors are and who influences the system. Furthermore, the innovation advisor needs to be able to connect to the community. Therefore, he/she needs to understand his/her own role in the system. Understanding the broader network will allow the innovation advisor to identify relevant actors.

### 4.2.2. Understanding the AKIS

The second cluster of the theme 'content competence' also concerns an understanding of the project context; however, this cluster focusses more specifically on the local 'Agricultural Knowledge and Innovation system' (AKIS), a concept that describes knowledge exchange in a certain region and the services supporting this exchange [43]. It is about understanding the main actors in this system, as well as the political and economic context. A prerequisite for this competency is basic knowledge about legal matters and the public policy of the region.

### 4.2.3. Content Knowledge

The third cluster of the theme 'content competence' involves several basic requirements that make it possible for the innovation advisor to function within the context of the innovation project. In terms of background and training, it was deemed beneficiary, although not essential, for the innovation advisor to have a background in agriculture and have a certain degree of technical knowledge. Importantly, this is not only about the technical knowledge itself but also about relating to the actors in the project and gaining their trust. One of the interviewees mentioned, from his own experience, the importance of informing yourself about the sector in which the actors are operating: "*I always make sure that I look through the trade journals in advance and that I know the price of tomatoes and pork. When it comes up, I can reply 'the prices haven't been that good lately'. Or on the contrary 'it will be difficult to avoid the taxes'. That usually helps to have a conversation.*"

This technical knowledge is not always necessary and depends on the project the innovation advisor works on. One of the interviewees pointed out that it might be worthwhile for the innovation advisor to possess T-shaped skills, i.e., the advisor has one area of expertise as well as broad knowledge in other areas.

A second function of technical knowledge is to help the innovation advisor access the knowledge needed by the group, as well as where to access that knowledge. On a related note, it was deemed useful but not essential for the innovation advisor to be able to speak English, as proficiency in English will help the innovation advisor to access knowledge (research papers, literature, trainings). This was not essential as in some cases this knowledge may be available in the native language.

### *4.3. Methodological Competence*

The theme 'methodological competence' comprises four clusters which are specifically related to the context of Interactive Innovation: (a) understanding the innovation process, (b) energy, (c) co-creation, and (d) mediation.

### 4.3.1. Understanding the Innovation Process

The first cluster of the theme 'methodological competence' is specifically related to facilitating innovation processes. The innovation advisor should possess a certain sensitivity for the process, which makes it possible to recognise patterns in an innovation process and to know whom to mobilise in what stage of the process. This also implies that an innovation advisor should know how to act in any given situation and be able to choose

appropriate actions. Furthermore, the innovation advisor needs tools related to innovation processes to monitor if the group is still on track. An example of such a qualitative tool is learning histories, which can help other actors to obtain a better view of the situation. This cluster also concerns the problem-solving skills the innovation advisor should possess.

### 4.3.2. Energy

The second cluster of the theme 'methodological competence' focusses on being able to keep energy and enthusiasm in the group. One of the interviewees commented: "*Very often in such processes you see that if you manage to bring different parties together, there is initially a lot of enthusiasm: 'Hey, it's great that we are meeting and working on this solution'. But then you see that it is difficult to keep the energy in the group.*"

The innovation advisor needs to be able to activate and mobilise the actors in the network. Furthermore, the innovation advisor needs to moderate the group and thus needs facilitation skills.

To enable a common understanding, actors coming from different backgrounds and with a different area of expertise need to understand each other, therefore the innovation advisor needs translation skills. Talking about this issue, one of the interviewees told an anecdote about a meeting between farmers and researchers he had organised: " ... *after the meeting they (the farmers) said, 'Thank you, it was very entertaining, but to be honest, we don't understand this'. . . . So how to solve this problem of the different languages? The practical language of farmers, and the theoretical language of researchers. . . . That is the role for advisors, to be in the middle.*"

### 4.3.3. Co-Creation

The cluster of 'co-creation' relates to the recognition of the crucial positions in the network, as well as being able to identify missing positions in the group. The innovation advisor should also possess a good insight into human psychology to be able to fill the missing positions.

### 4.3.4. Mediation

The last cluster includes the necessary skills for mediation. Conflicts can arise in groups, and it is the role of the innovation advisor to respond adequately.

### *4.4. Organisational Competence*

All competencies identified under theme 'organisational competence' were grouped into a single cluster under the same name. This cluster includes the practical network management skills that an innovation advisor should possess. This includes skills such as planning, organising meetings, following up with contacts, keeping track of the network, time management, resource management, etc. It was also deemed useful for the innovation advisor to be able to write a project proposal and to know how to collect funds, although this was not considered to be essential. Delegation skills ease the organisational burden on the innovation advisor. Furthermore, basic digital skills are seen as essential for carrying out organisational tasks, as well as accessing new information.

### *4.5. Reflection, Learning and Personal Development*

Professionals are expected to constantly be improving the quality of their work. Four clusters were identified under theme 'reflection, learning, and personal development'; they are (a) reflection among peers, (b) self-reflection, (c) addressing professional network, and (d) lifelong learning.

### 4.5.1. Reflection among Peers

The first cluster is about peer evaluation. Innovation advisors should have the habit of reflecting upon their work with their peers. This requires sharing a common language. One interviewee said: "*Reflection among peers is much more effective then personal reflection*

*alone. But for this reflection it is really important people have a common language. Handling the common language is the skill. That is something you can learn by doing.*"

### 4.5.2. Self-Reflection

The second cluster concerns the individual act of self-reflection. Complementary to reflecting among peers, innovation advisors should habitually reflect on their work on their own.

### 4.5.3. Addressing Professional Network

This cluster implies that the innovation advisor is able to call upon a professional network. This competency relies on the organisational skill of keeping track of the network but goes a step further: Addressing the network implies knowing who to approach in a specific situation. One of the interviewees commented: "*If the project is about pigs, then it is expected of you as an intermediary that you have a good network and that you know who has something to say about pigs. Not because you are going to tell it yourself, but because you know how to approach them for the questions that play a role in your project.*"

### 4.5.4. Lifelong Learning Aptitude

The fourth and last cluster deals with competency development and learning. The innovation advisor should have the habit of learning, taking in new experiences, and knowing how to find new information. This includes actively seeking out opportunities for training.

This cluster is linked to the first three clusters of reflection, learning, and personal development. The elements in these clusters allow the innovation advisor to undertake lifelong learning and build competencies in the various areas.

## 5. Discussion

### 5.1. Reflections: Towards Collective Competence

If agricultural advisors are expected to play the role of intermediary in innovation processes, the competencies required to do this successfully must be clearly defined. The competency profile presented in this article represents a possible first step. The identified competencies include skill sets, attitudes, values, personal traits, habits, knowledge, etc. In accordance with Mulder's (2001, 2007) definition, we consider these characteristics (i.e., competencies) to be the building blocks of a person's general competence [26]. Competence, in turn, whether stemming from a single person or a team of people, generates a certain capability.

To what extent does this competency profile coincide with the current profile of a (technical) agricultural advisor? In contrast to the academic consensus that agricultural advisors also need to be skilled as educators and facilitators of learning, it appears that most advisors primarily rely on their technical qualifications [13]. Generally, advisors are required to have a minimum of a bachelor's degree in agricultural science or other related sciences and, for subject matter experts, Master's degrees are even more common [13,44]. Other competencies needed to perform advisory work are *learned by doing* over time [13,45]. These skills and knowledge coincide with the competencies belonging to the second theme 'content competence' and to the fourth theme 'organisational competence'.

The first theme, 'basic disposition and attitude', requires a certain personal-level commitment of the advisor to the job being performed. Literature reports that this trait appears to be generally present among agricultural advisors [11,19]. Above that commitment, however, the first theme also emphasises empathy, emotional intelligence, reliability, and professional ethics. It is often assumed that these attributes largely depend on the nature of the person in question; the extent to which these attributes can be taught and learned forms a potential topic for further research.

The third theme, 'methodological competence', comprises competencies of a different register than those associated with the technical background of the agricultural advisor.

This theme requires certain social skills which are needed for guiding a group of people and is thus directly related to the co-creation and multi-actor aspect of Interactive Innovation.

The last theme 'reflection, learning, and personal development', requires the advisor to possess the commitment and habit of reflecting on his or her work, connecting with a professional network, and seeking out training activities. Gorman (2019) shows that critical reflection and dialogue are necessary to process learning from experience. She stresses the importance of different levels of reflection in higher education aimed at agricultural advisory services.

It thus appears that most of the competencies presented in this profile are not unique to innovation support. Some competencies, presented in this profile, are also needed to perform other tasks advisors are expected to perform. However, the need to combine a diversity of competencies might be unique to innovation support, as we are expecting people with a technical background to take up a role where their technical knowledge serves as a base but not as their primary skill set. This technical knowledge needs to take part in an interplay of diverse competencies in order to successfully bring about co-creation between multiple stakeholders.

At present, agricultural advisors usually start their education with a focus on technical skills, which forms a starting point from which other competencies are acquired through 'learning by doing' on the job. An alternative could be a training with a focus on the intermediary role. This begs the question, "Do we need to focus on the competence development of individuals or should more people be involved in the innovation process as a team?"

As mentioned previously, innovation service provision is diverse [46]. The provision of different ISS is therefore likely to require a diversity of capabilities and thus different people possessing different competencies. For example, 'Networking, facilitation and brokerage' services require methodological competencies, while 'Enhancing/supporting access to resources' services require certain content competences (e.g., understanding social environment, understanding the AKIS) as well as organisational competencies (e.g., managing resources, knowing how to collect funds). It seems unrealistic to expect many individuals to possess the complete range of competencies required. Instead of relying on a single person to provide both types of services, a project might mobilise different actors with complementary competencies.

This emphasis on collective skills versus individual skill sets was also mentioned in Albaladejo et al. (2007) in their interpretation of several oral and written accounts of agricultural advisors regarding new skills required of them [11]. One of the innovation intermediaries claimed to have a feeling of having to be an octopus with eight arms and do everything [11]. The competency profile presented here may be used in two ways: (1) as a tool to reflect on the attributes needed in the context of a specific Interactive Innovation project or in a specific stage of the innovation project and (2) as a guide when searching for a team with complementary skills that can cover all the competencies required for the project.

When assembling such a complementary team, the concept of a competence broker could be useful. This concept is typically used in a business innovation context [47], but it might also be applicable in the context of agricultural innovation systems. On an organisational level, the competence broker collects available knowledge about existing competence within a company, directs competence exchange where needed, and provides such direction even where it has not been asked for [47]. Scaling this broker role to the level of the AKIS presents great potential in the current privatised and pluralistic landscape [6]. In a context of complex interests and power relations [6], a competence broker could act as a facilitator by matching certain actors or organisations on the basis of their 'core competencies' in the context of a specific project. The concept of core competencies was first used as corporate strategy by Prahalad & Hamel (1990). It describes the main strength of an organisation that can be converted into a variety of products or services [48]. The competency profile presented in this article could serve as a point of departure or even as

an intervention tool for the competence broker. Before we would be able to advise on how best to use the profile, further work will be required to fully understand the interaction and cooperation of several actors possessing different sets of competencies.

Another practical application of the competency profile is the potential for using it to aid design of new training materials. When the profile was being created, some questions regarding the trainability of the competencies arose. It appeared some competencies do not seem to be as straightforward to train as others. Further research could usefully explore how to create an environment, which will stimulate the acquisition of these harder to train competencies.

*5.2. Limitations*

Apart from the literature review, the result of this research was based on a limited number of interviews ($n = 7$) with a small yet diverse group of experts. Even after a modest number of interviews, the data collection reached a point of saturation.

A second limitation is the fact that this study neglects the context dependency of the competencies [27,31]. As mentioned earlier, we view the value of this profile as a tool of reflection or intervention during the set-up of an innovation project. Reflecting on the specific context of the project will bring the required competencies to the fore and help in assembling a team that possesses the right capabilities.

## 6. Conclusions

To conclude, this article responds to the changing role of the agricultural advisor into the role of "innovation intermediary" by developing a clear competency profile for that new role. The profile was based on a literature study and interviews with experts in co-creation processes and was validated by members of the i2connect consortium.

The sheer number and diversity of competencies listed in this profile indicate that it might not be reasonable to expect a single person to fulfil these expectations. We therefore propose that a group of people should be assembled who collectively possess the required capabilities, as this complementary team would be better suited to handle the complexity of an innovation process.

The competency profile presented in this article can serve as a tool to compose such a team. However, further research is needed to give advice on how the profile can be used to identify which functions are needed for a specific innovation project and how to form a team of people with the capability to perform each function.

Further research might also explore to what extent an individual can acquire these competencies and how to create an environment that will stimulate the acquisition of harder to train competencies. Limitations of this study are that the results are based on a limited number of interviews and that this research does not consider the context dependency of the competencies.

**Author Contributions:** Conceptualization, C.L. and L.D.; methodology, C.L. and L.D.; validation, C.L., L.D., E.K. and F.M.; formal analysis, C.L.; investigation, C.L.; data curation, C.L.; writing—original draft preparation, C.L.; writing—review and editing, C.L., L.D., E.K. and F.M.; visualization, C.L.; supervision, L.D., E.K. and F.M.; project administration, L.D.; funding acquisition, L.D. and F.M. All authors have read and agreed to the published version of the manuscript.

**Funding:** This research was conducted within the i2connect project. This project was funded by the European Union's Horizon 2020 research and innovation programme under grant agreement No 863039.

**Institutional Review Board Statement:** This study followed the ethic requirements of the i2connect project, which were established in accordance with the European Commission, grant agreement ID: 863039.

**Informed Consent Statement:** Informed consent was obtained from all subjects involved in the study.

**Data Availability Statement:** The data presented in this study are available on request from the corresponding author. The data are not publicly available due to privacy restrictions.

**Acknowledgments:** The authors would like to thank the participants in the interviews and validation workshop for their contribution, Miriam Levenson for her English revisions, and the reviewers for their useful comments on the manuscript.

**Conflicts of Interest:** The authors declare no conflict of interest. The funders had no role in the design of the study; in the collection, analyses, or interpretation of data; in the writing of the manuscript, or in the decision to publish the results.

## Appendix A

**Table A1.** Coding tree.

| **Basic disposition and attitude** | |
|---|---|
| | Accountability |
| | Action skills |
| | Being creative |
| | Being sociable |
| | Critical thinking |
| | Curious attitude and broad interest |
| | Dedication, passion |
| | Empathy and emotional intelligence |
| | Ethics |
| | Flexible and easy going |
| | Open mindedness |
| | Problem solving |
| | Responsibility and professional attitude |
| | Self-awareness |
| | Sensitivity |
| | Trust in intuition |
| | Trustworthiness |
| **Content competence** | |
| | Ability to speak English |
| | Agricultural, technical knowledge |
| | Basic knowledge about public policy, legal matters, copyrights, and organisational structure |
| | Educational background |
| | Good overview of situation |
| | Process sensitivity |
| **Methodological competence** | |
| | Being able to keep energy and enthusiasm in the group (motivational skills) |
| | Being able to moderate, facilitation skills |
| | Communication skills |
| | Educational and teaching skills |

**Table A1.** *Cont.*

|  |  |
|---|---|
|  | Insight in human psychology |
|  | Mediation skills |
|  | Mobilisation skills |
|  | Networking skills |
|  | Translation skills |
| **Managerial and organisational competence** |  |
|  | Background and neutral position |
|  | Management and organisational skills |
|  | Mobility |
| **Reflection, learning, and personal development** |  |
|  | Ability to find way to new information |
|  | Personal development |
|  | Professional network |
|  | Reflection |

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
