# Peer review of "Competencies for Agricultural Advisors in Innovation Support"

_sustainability, doi:10.3390/su14010182_

Round 1

Reviewer 1 Report

Thank you very much for your interesting paper. This paper adequately combines a review of the literature with case studies to build the figure of agricultural advisors. It is necessary to highlight the conclusion in which it is stated that teamwork contributes more to the company than individual work.

Reviewer 2 Report

Dear authors and editor,

the manuscript “Competencies for agricultural advisors in innovation support” aims at defining a competency profile for agricultural advisors in facilitating Interactive Innovation. To highlight the skills, the authors used literature review, interviews, and workshops. The topic seems engaging in a changing agricultural world. Some remarks to this work follow:

  • In the Introduction, your presentation is very interesting. However, it sounds more like a sociological work than an agricultural one. I suggest introducing a section relating to the recent agricultural innovations and linking them to sustainability. This would help best meet the scopes of the Journal.
  • In the Material and Methods section, more information is needed about the literature search. Where did you get the papers from? What did you look for (provide the search string)? How did you analyse the abstract and keywords? Moreover, information is needed about the clustering methodology.
  • English must be deeply revised. Check for mistakes and syntax.

Here are some specific comments:

Line 26: check for mistakes

Lines 107, 175: why did you use italics for “and”?

Line123-126: this sentence is too long and hard to read

Line 146: advisory must not be capital

Table 1: what’s AKIS? I had to search on google

Line 23: check for mistakes

Reviewer 3 Report

Dear Authors,

Following are some comments:

1) Introduction section needs to be restructured.

  • First paragraph - Establishing a territory (claiming centrality; making topic generalisation; reviewing items of previous research)
  • Second paragraph - Establishing a niche (counter-claiming; indicating a gap; question raising; continuing a tradition)
  • Third paragraph - occupying the niche (outlining purposes; announcing present research; announcing principal findings; indicating structure)
  • Also, you can reference this article "COVID-19 demand-induced scarcity effects on nutrition and environment: investigating mitigation strategies for eggs and wheat flour in the United Kingdom"

2. Conclusion section needs to highlight the limitations and recommendations

3. Author contributions: Use initials rather than full names. For example, Charlotte Lybaert (C.L.)

Overall, its a good piece of work. Good luck!
